# Global, regional, and national burden and trends of kidney cancer associated with high BMI from 1990 to 2021: Findings from the Global Burden of Disease Study 2021

Hong Luo⊚, Hailiang Jing⊚, Houyu Zhao, Yun Zhao⊚*

Department of Oncology, Yancheng Branch of Nanjing Drum Tower Hospital, Yancheng Second People's Hospital, Yancheng, Jiangsu Province, China

⊚ These authors contributed equally to this work.
* 13770000845@163.com

## Abstract

### Background

Kidney cancer represents a significant health concern that profoundly impacts the well-being of individuals, particularly those with a higher Body Mass Index (BMI). Although kidney cancer's impact is substantial, there remains a paucity of comprehensive global research dedicated to elucidating the specific burden attributable to kidney cancer in association with elevated BMI levels. This study endeavors to address this gap by examining the global distribution, incidence rates, and disability-adjusted life years (DALYs) linked to kidney cancer as a consequence of high BMI between the years 1990 and 2021. Utilizing data sourced from the Global Burden of Disease (GBD) 2021 study, the research seeks to provide a clearer understanding of how excess body weight contributes to the global kidney cancer burden.

### Methods

The methods utilized in our comprehensive analysis were grounded in kidney cancer data sourced from the Global Burden of Disease (GBD) 2021 report. This data was meticulously examined to understand the distribution, incidence rates, and disability-adjusted life years (DALYs) pertaining to kidney cancer across 204 countries and regions. The information was stratified by age group, sex, calendar year, geographical area, and Socio-demographic Index (SDI) to provide a detailed overview of the disease's impact. To evaluate temporal trends and shifts within these metrics, we employed the Estimated Annual Percentage Change (EAPC) methodology, thereby allowing for a nuanced assessment of how kidney cancer patterns have evolved over the period studied.

### Results

From 1990 to 2021, the global incidence of kidney cancer associated with high Body Mass Index (BMI) witnessed a substantial increase. By 2021, it was estimated that there

**Data availability statement:** All data are available from the GBD database (https://vizhub.healthdata.org/gbd-results/).

**Funding:** The author(s) received no specific funding for this work.

**Competing interests:** The authors have declared that no competing interests exist.

were approximately 720,000 new cases, a significant rise from the roughly 500,000 cases recorded in 1990. Over this period, global trends indicated rising age-standardized incidence rates (ASIRs) and disability-adjusted life years (DALY) rates for kidney cancer. The Estimated Annual Percentage Change (EAPC) for both ASIR and DALY rate was positive, indicating an upward trend in kidney cancer's global impact. Regions characterized by middle Socio-demographic Index (SDI) levels reported the highest absolute numbers of kidney cancer cases, whereas areas with high SDI levels demonstrated the highest per capita rates. Incidence rates were found to peak among middle-aged individuals. Notably, males experienced higher rates of kidney cancer compared to females across all age brackets, suggesting a gender disparity in the disease's prevalence. These findings underscore the need for targeted interventions and public health strategies aimed at addressing the growing burden of kidney cancer, particularly in populations with high BMI.

## Conclusion

The global impact of kidney cancer associated with high BMI has expanded notably from 1990 to 2021, highlighting significant variations across different SDI regions, countries, and sexes. This increasing trend underscores the need for targeted interventions and public health strategies, particularly in regions and populations where kidney cancer prevalence is disproportionately high due to high BMI. Strengthening preventive measures and raising awareness about the risks of high BMI could help mitigate the growing burden of kidney cancer worldwide.

## Introduction

High Body Mass Index (BMI) is increasingly recognized as a significant risk factor leading to various health issues, including kidney cancer [1]. Obesity affects individuals across all age groups; however, its prevalence and associated health risks are particularly notable among younger populations, impacting their overall quality of life. This period is crucial for education, career establishment, and social interactions [2]. The long-term effects of a high Body Mass Index (BMI), including an elevated risk of comorbidities and diminished quality of life, underscore the significance of tackling this issue proactively [3].

Previous studies have reported obesity and a high BMI to be risk factors for kidney cancer incidences and mortality [4]. Understanding the epidemiological characteristics of kidney cancer in relation to high BMI can aid in implementing early detection, prevention, and management strategies to alleviate the long-term burden of the condition and improve outcomes for this vulnerable group [5]. In recent years, kidney cancer has become a major contributor to global disability-adjusted life years (DALYs), with a significant impact on the population. Kidney cancer incidence and mortality rates are on the rise [6]. This condition often manifests as recurring health problems that can severely disrupt daily activities. Symptoms may include pain, fatigue, and other systemic disturbances. Multiple factors can lead to the development of kidney cancer, with lifestyle choices such as diet and physical activity playing a key role [7]. Despite its rising incidence, kidney cancer often does not receive sufficient attention or treatment, especially in low- and middle-income countries, where research, education, and clinical resources are limited [8].

While existing research has examined the global impact of high BMI on health outcomes, there is a notable gap in studies specifically addressing kidney cancer. Our study aims to provide the most current and comprehensive analysis of kidney cancer epidemiology by utilizing

the latest data from the Global Burden of Disease (GBD) 2021 study and employing advanced statistical methods, such as Estimated Annual Percentage Change (EAPC) projections. These methodologies allow us to analyze temporal trends, untangle the relative contributions of various population factors, and offer a forward-looking perspective on the burden of kidney cancer. Additionally, our study captures the potential impact of recent global events, such as the COVID-19 pandemic, on the burden of kidney cancer. This provides insights into how recent global health crises might have influenced the prevalence, incidence, and DALYs associated with kidney cancer. By addressing these gaps and incorporating contemporary data, our research hopes to contribute valuable knowledge that can inform policy decisions and guide future healthcare initiatives aimed at reducing the incidence and improving the outcomes of kidney cancer globally.

## Methods

### Data collection and processing

For this study, we sourced data on kidney cancer associated with high Body Mass Index (BMI) from the Global Burden of Disease (GBD) 2021 database, a comprehensive repository known for providing the latest estimations on a wide array of diseases, injuries, and risk factors across numerous countries and territories, organized into broader regions. Utilizing the GBD 2021 framework, which segments the globe into 21 distinct geographic regions based on epidemiological patterns and geographic adjacency, allowed us to gain deeper insights into the variability of disease burden across different parts of the world. This systematic regional classification includes diverse areas such as Andean Latin America, Australasia, the Caribbean, Central Asia, Central Europe, and others. This categorization has been consistently employed in previous GBD studies and has proven effective in analyzing and comparing health metrics across geographically and epidemiologically varied regions.

In our analysis, we focused on extracting kidney cancer data relevant to prevalence, incidence, and Disability-Adjusted Life Years (DALYs). These metrics were complemented with their respective uncertainty intervals (UI), providing a measure of statistical reliability. The calculation of DALYs took into account both years lived with disability (YLD) and years of life lost (YLL), thus offering a holistic perspective on the burden of the disease. For a detailed methodology, please refer to the Supplementary Methods section.

To contextualize our findings within a broader framework, we employed the Socio-Demographic Index (SDI), a composite indicator that assesses regional development through income, education, and fertility metrics. The GBD 2021 divided countries and territories into five SDI tiers, ranging from low to high development statuses. This approach facilitated a nuanced understanding of how socioeconomic factors influence the incidence and progression of kidney cancer across different settings. All data utilized in this study are publicly available through the Global Health Data Exchange platform (https://ghdx.healthdata.org/gbd-2021/sources). The Institutional Review Board at the University of Washington has exempted GBD studies from requiring informed consent, ensuring that our research complies with ethical standards. Furthermore, our research adheres to the STROCSS reporting standards, guaranteeing transparency and rigor in our methodology and findings.

### Statistical analysis

To analyze the trends in age-standardized rates (ASR) of kidney cancer incidence, mortality, DALYs, and prevalence, we utilized the Estimated Annual Percentage Change (EAPC) method. This method involves constructing a regression model to represent the trend changes [9]:

$$ln(ASR) = \alpha + \beta X + e ln(ASR) = \alpha + \beta X + e \qquad [1]$$

where ln(ASR) represents the natural logarithm of the age-standardized rate, X denotes the calendar year, α is the intercept, β represents the slope indicating the trend over time, and e represents the error in the model. The EAPC is defined as $100 \times [\exp(\beta) - 1]$, indicating the annual percentage change [10]. Trends were analyzed using the EAPC and its corresponding 95% confidence interval (CI) [11]. If both the EAPC and the lower limit of the CI were positive, the trend was considered increasing; if both were negative, the trend was considered decreasing; otherwise, the trend was deemed stable [12]. We also used Gaussian process regression and Pearson correlation coefficient to explore the relationships between EAPC, age-standardized rates, and the Socio-Demographic Index (SDI). Additionally, we applied decomposition methods to analyze the changes in kidney cancer incidence, prevalence, and DALYs due to alterations in population age structure, growth, and epidemiological shifts. Furthermore, we employed the World Health Organization's Health Equity Assessment Toolkit for equity-focused analyses and used R statistical software for data processing and statistical computations.

## Results

### Global trends in disease incidence and mortality

The global burden of kidney cancer has garnered significant attention due to its increasing incidence and the associated health implications [13]. Understanding the patterns of kidney cancer incidence and mortality is crucial for developing effective public health strategies. According to the data summarized in Table 1, a mixed pattern in disease incidence and mortality has emerged on a global scale. While many regions experienced a decline in both the number of cases and age-standardized rates (ASRs) from 1990 to 2021, several others faced increasing trends. For instance, high-middle SDI areas observed a rise in the number of kidney cancer cases from 108,583 (with a 95% uncertainty interval of 42,626-177,410) in 1990 to 253,443 (with a 95% uncertainty interval of 102,935-411,349) in 2021. This increase was accompanied by a rise in the age-standardized rates (ASRs) from 10.51 (with a 95% uncertainty interval of 4.12-17.16) per 100,000 population in 1990 to 12.88 (with a 95% uncertainty interval of 5.23-20.9) per 100,000 population in 2021 (Table 1). This upward trajectory is reflected in the positive Estimated Annual Percentage Change (EAPC) value of 0.54 (with a 95% uncertainty interval of -0.41 to 0.68), indicating a growing disease burden in these regions. The positive EAPC value underscores the increasing trend in kidney cancer incidence, highlighting the need for targeted interventions to address the rising health challenge in high-middle SDI areas. Such interventions should aim to reduce risk factors, improve early detection, and enhance treatment options to mitigate the impact of kidney cancer on public health.

### Regional disparities in disease burden

Many diseases exhibit significant variations across different regions [14]. The information shows significant disparities in disease burden among different regions. High-income countries generally exhibit lower disease incidence and mortality rates than their low- and middle-income counterparts. For instance, Europe & Central Asia - WB reported fewer cases at 177,747 (69,696-289,655) in 1990 and 310,105 (127,208-498,996) in 2021, with corresponding ASRs of 16.87 (6.61-27.47) and 20.08 (8.28-33.31) (Table 1). In contrast, low SDI regions like Africa and South Asia faced higher disease burdens, with Africa reporting 5,546 (2,082-8,730) cases in 1990 and 29,093 (11,504-46,409) in 2021, along with ASRs of 1.77 (0.69-2.78) and 3.99 (1.57-6.34) (Table 1).

**Table 1. Global trends in disease incidence and mortality (1990-2021).**

| Location | 1990 | | 2021 | | EAPC (95% CI) |
|---|---|---|---|---|---|
| | Number (95% UI) | ASR (95% UI) | Number (95% UI) | ASR (95% UI) | |
| **High-middle SDI** | 108583 (42626-177410) | 10.51 (4.12-17.16) | 253443 (102935-411349) | 12.88 (5.23-20.9) | 0.54 (0.41-0.68) |
| **High SDI** | 168243 (66400-273509) | 15.65 (6.18-25.42) | 312529 (129704-498459) | 16.15 (6.7-25.64) | 0.08 (-0.04-0.21) |
| **Low-middle SDI** | 8458 (3283-13699) | 1.24 (0.48-2.02) | 48210 (19117-78003) | 3.1 (1.23-5.03) | 3.19 (3.11-3.27) |
| **Low SDI** | 2506 (939-4092) | 0.99 (0.37-1.62) | 11482 (4012-19353) | 1.99 (0.7-3.33) | 2.27 (2.14-2.41) |
| **Middle SDI** | 29659 (11466-48316) | 2.57 (0.99-4.2) | 154799 (63282-253631) | 5.5 (2.25-9.03) | 2.5 (2.46-2.53) |
| **Africa** | 5546 (2082-8730) | 1.77 (0.66-2.78) | 29093 (11504-46409) | 3.99 (1.57-6.34) | 2.78 (2.71-2.84) |
| **America** | 95336 (38081-154061) | 15.63 (6.25-25.26) | 243260 (100656-380733) | 18.47 (7.64-28.89) | 0.47 (0.36-0.57) |
| **Asia** | 41144 (15934-67226) | 1.88 (0.73-3.07) | 204672 (79288-340598) | 3.95 (1.53-6.57) | 2.48 (2.39-2.57) |
| **Australasia** | 3582 (1375-5921) | 15.5 (5.96-25.58) | 8820 (3632-14138) | 17.79 (7.29-28.35) | 0.45 (0.34-0.57) |
| **Europe** | 175013 (68666-285079) | 17.11 (6.71-27.84) | 302492 (124092-486754) | 20.35 (8.35-32.74) | 0.5 (0.36-0.63) |
| **Europe & Central Asia - WB** | 177747 (69696-289655) | 16.87 (6.61-27.47) | 310105 (127208-498996) | 20.08 (8.24-32.31) | 0.49 (0.35-0.63) |
| **High-income Asia Pacific** | 7316 (2784-11962) | 3.53 (1.34-5.78) | 19774 (7479-32089) | 4.74 (1.8-7.6) | 0.94 (0.74-1.15) |
| **High-income North America** | 67586 (27173-109468) | 20.57 (8.28-33.31) | 129877 (54728-201168) | 21.08 (8.89-32.52) | -0.02 (-0.21-0.18) |
| **South-East Asia Region** | 5420 (2007-8821) | 0.67 (0.25-1.09) | 35685 (13214-57909) | 1.82 (0.67-2.97) | 3.13 (3.01-3.25) |
| **South Asia** | 3396 (1265-5504) | 0.53 (0.2-0.85) | 25772 (9506-41702) | 1.62 (0.6-2.63) | 3.73 (3.67-3.79) |
| **Western Africa** | 813 (286-1372) | 0.96 (0.34-1.61) | 4492 (1632-7350) | 2.3 (0.84-3.76) | 3.03 (2.96-3.1) |
| **Western Europe** | 89943 (35484-146063) | 16.38 (6.47-26.54) | 142947 (57188-233539) | 16.73 (6.7-27.28) | 0.12 (0.05-0.2) |
| **American Samoa** | 1 (1-2) | 5.2 (1.96-8.63) | 4 (2-6) | 7.68 (3.33-11.92) | 1.61 (1.29-1.93) |
| **Australia** | 2967 (1126-4902) | 15.38 (5.84-25.37) | 7323 (3009-11814) | 17.52 (7.16-28.06) | 0.42 (0.29-0.55) |
| **Canada** | 4714 (1841-7806) | 14.8 (5.79-24.46) | 11562 (4715-18616) | 17.08 (6.99-27.39) | 0.74 (0.49-0.99) |
| **China** | 16154 (6380-26922) | 1.76 (0.69-2.95) | 96676 (35994-170318) | 4.62 (1.72-8.17) | 3.52 (3.32-3.73) |

## Influence of socioeconomic factors on health outcomes

Socioeconomic factors, including education level, income, employment status, and housing conditions, can significantly influence the health outcomes of individuals and populations [15]. The information highlights the relationship between socioeconomic status and disease burden, illustrating that regions with higher socioeconomic standing generally maintain lower disease burdens compared to those with lesser status. High-income Asia Pacific countries, for example, recorded 7,316 cases (2,784-11,962) in 1990, increasing to 19,774 cases (7,479-32,089) by 2021, corresponding to age-standardized rates (ASRs) of 3.53 (1.34-5.78) and 4.74 (1.87-8.63), respectively (Table 1). Progress towards reduction was modest, as indicated by an estimated annual percentage change (EAPC) of -0.02 (-0.21-0.18) (Table 1). In low Socio-demographic Index (SDI) regions such as parts of Africa and South Asia, despite substantial resource constraints, significant improvements were observed in managing disease burdens. Africa, specifically, saw a rise in case numbers from 5,546 (2,082-8,730) in 1990 to 29,093 (11,504-46,409) in 2021, yet the ASRs declined from 1.77 (0.69-2.78) to 3.99 (1.57-6.34) (Table 1). This trend is reflected in a positive EAPC value of 2.78 (2.71-2.84) (Table 1), indicating the effective application of preventive measures and better access to healthcare services. The data provide a nuanced understanding of global disease trends and highlight the complex inter-play between socioeconomic conditions and health outcomes. Analysis of these specific data points reveals patterns and disparities that can inform evidence-based policy decisions aimed at reducing disease burdens. It emphasizes the necessity for ongoing investment in public health, regionally adapted interventions, and coordinated efforts among various stakeholders to achieve further reductions in disease burdens globally (Table 1).

## Age and regional distribution of deaths, DALYs, and cases

Upon examining the charts presented in Fig 1, a comprehensive overview of the distribution of deaths, Disability-Adjusted Life Years (DALYs), and the number of cases across multiple age groups and regions becomes evident. A striking observation is the elevated mortality rate observed among individuals aged 80 years and over, which correlates with the highest concentration of DALYs within this demographic. This indicates a substantial effect on both life expectancy and the quality of life for the elderly population. Interestingly, Graph B illustrates a higher incidence of cases within younger age brackets; however, the associated DALY count remains relatively low. This suggests that while the prevalence of the condition is more common among younger individuals, the severity and overall impact on their health and well-being appear to be less pronounced. Further insights are gleaned from Graphs C and D, which underscore the regional disparities in death rates and DALYs. These variations highlight the necessity for tailored interventions and strategic resource allocation to regions bearing a heavier disease burden. The data encapsulated within these charts offer pivotal information on the epidemiological landscape, equipping researchers and policymakers with critical knowledge to inform decisions and develop strategies aimed at addressing the health challenges faced by the most impacted regions and populations.

## Health disparities illustrated: Socioeconomic and gender perspectives

Health disparities refer to differences in the presence of disease, health outcomes, or access to health care across different groups of people [16]. Gaining insight into the dynamics of health and disease within populations is essential for enhancing public health policies and practices. While we might be aware of the intricacies of healthcare and disease burden in our day-to-day lives, it's all too easy to miss the broader patterns that define these experiences on a global scale. As depicted in Fig 2, there is a vivid illustration of how disease impacts vary

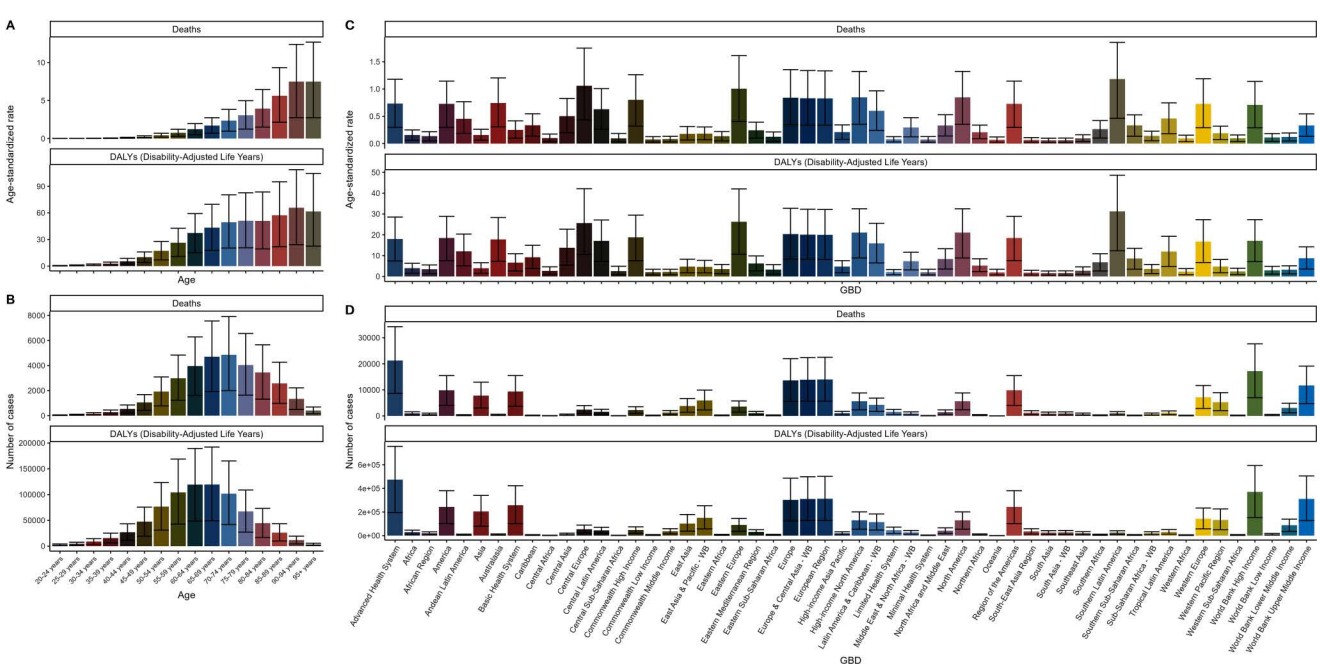

**Fig 1. Age and regional distribution of deaths, DALYs, and cases (1990-2021).**

significantly across different socioeconomic statuses and genders. In our daily observations, it's apparent that wealthier communities generally enjoy superior access to healthcare services and preventive measures. However, Fig 3 presents an unexpected revelation: regions with a higher sociodemographic index (SDI) actually exhibit higher age-standardized death rates and Disability-Adjusted Life Years (DALYs) compared to those with lower SDIs. This finding implies that despite possessing greater resources, more developed regions continue to grapple with considerable disease burdens. Considering the differences between men and women, societal roles and expectations invariably come to the forefront. Fig 3 accentuates a significant disparity, showing that males face higher death rates and DALYs than females. This discrepancy might be attributed to a myriad of factors, such as biological distinctions, increased exposure to occupational hazards, or cultural attitudes towards seeking medical assistance. In summary, Fig 3 emphasizes that comprehending health disparities demands a multifaceted perspective. It highlights the necessity for targeted interventions and equitable distribution of resources to tackle the specific challenges faced by diverse demographics. By acknowledging these patterns, we can strive to develop policies that foster healthier outcomes for all individuals, irrespective of their socioeconomic status or gender. This holistic approach is fundamental to advancing global health equity and ensuring that no one is left behind in the pursuit of improved health and well-being.

## Countries level

Understanding the dynamics of health and disease within populations is essential for enhancing public health policies and practices [17]. Comparing age-standardized mortality rates across different nations reveals varied outcomes, with some countries exhibiting higher rates than others. Notably, China maintains notably lower mortality rates than many Western counterparts; yet, a closer look at the data through graphical representations offers deeper insights. Fig 2, which illustrates standardized Disability-Adjusted Life Years (DALYs), reveals a distinctive trend within the Northern regions of China, where there has been a marked escalation in DALYs, indicating an increased morbidity burden despite the lower mortality

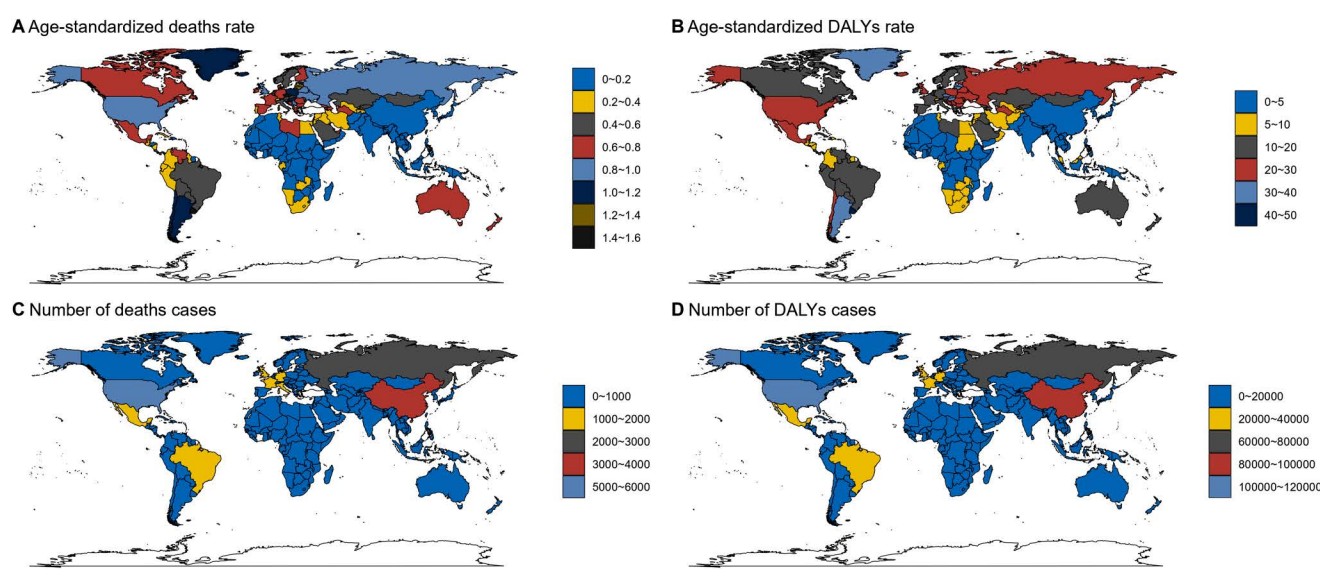

**Fig 2. Socioeconomic and gender perspectives.**

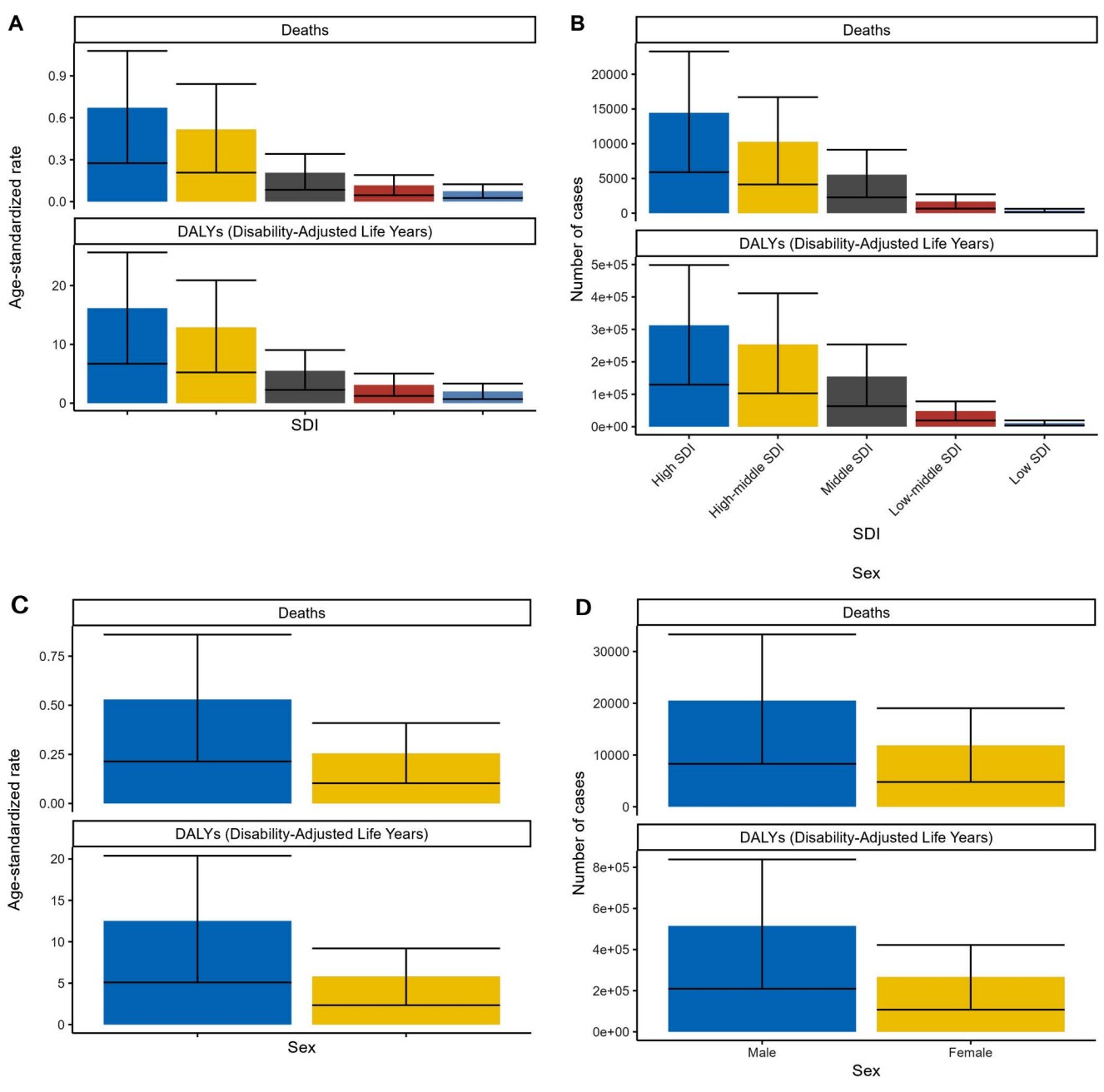

**Fig 3. Countries level.**

rate. Further analysis of (Fig 2C, 2D)highlights a clear association between elevated Body Mass Index (BMI) and the incidence of renal cell carcinoma (RCC). Epidemiological data suggest that RCC not only elevates mortality rates but also imposes a considerable toll in terms of DALYs, presenting a major public health concern. Given the context of China's socio-economic transformation, characterized by rapid urbanization and lifestyle changes, the rise of obesity-related conditions such as RCC poses a multifaceted challenge necessitating a comprehensive strategy that includes both preventive and curative measures, integrating policies promoting healthier lifestyles with enhanced medical interventions for early detection and

treatment, alongside tailoring public health initiatives to address the disparity in NCD burden across different provinces using evidence-based interventions and strengthened healthcare infrastructure to ensure equitable access to cancer screening and treatment services.

## Trend analysis of health burdens in china and neighboring countries

In our analysis (Fig 4A), it is noteworthy that the Estimated Annual Percentage Change (EAPC) in age-standardized death rates in China and its neighboring countries has demonstrated a significant increase compared to other nations. The graphical representation clearly delineates that the EAPC for age-standardized Disability-Adjusted Life Years (DALYs) follows a parallel trend, suggesting that the burden of disease, measured by DALYs, is also on the rise, impacting both quality of life and longevity (Fig 4A). Meanwhile (Fig 4B), Figure C indicates that the change in the number of death cases in China is moderate at a national level; however, a closer examination reveals a pronounced upward trajectory specifically in the northern and southwestern regions of the country, as well as in other neighboring nations. The map further reveals that the adjusted DALYs changes are notably high in China, aligning with the broader global context of rapid changes and increasing interconnectedness of health systems worldwide (Fig 4B). With the ongoing transformations driven by globalization, technological

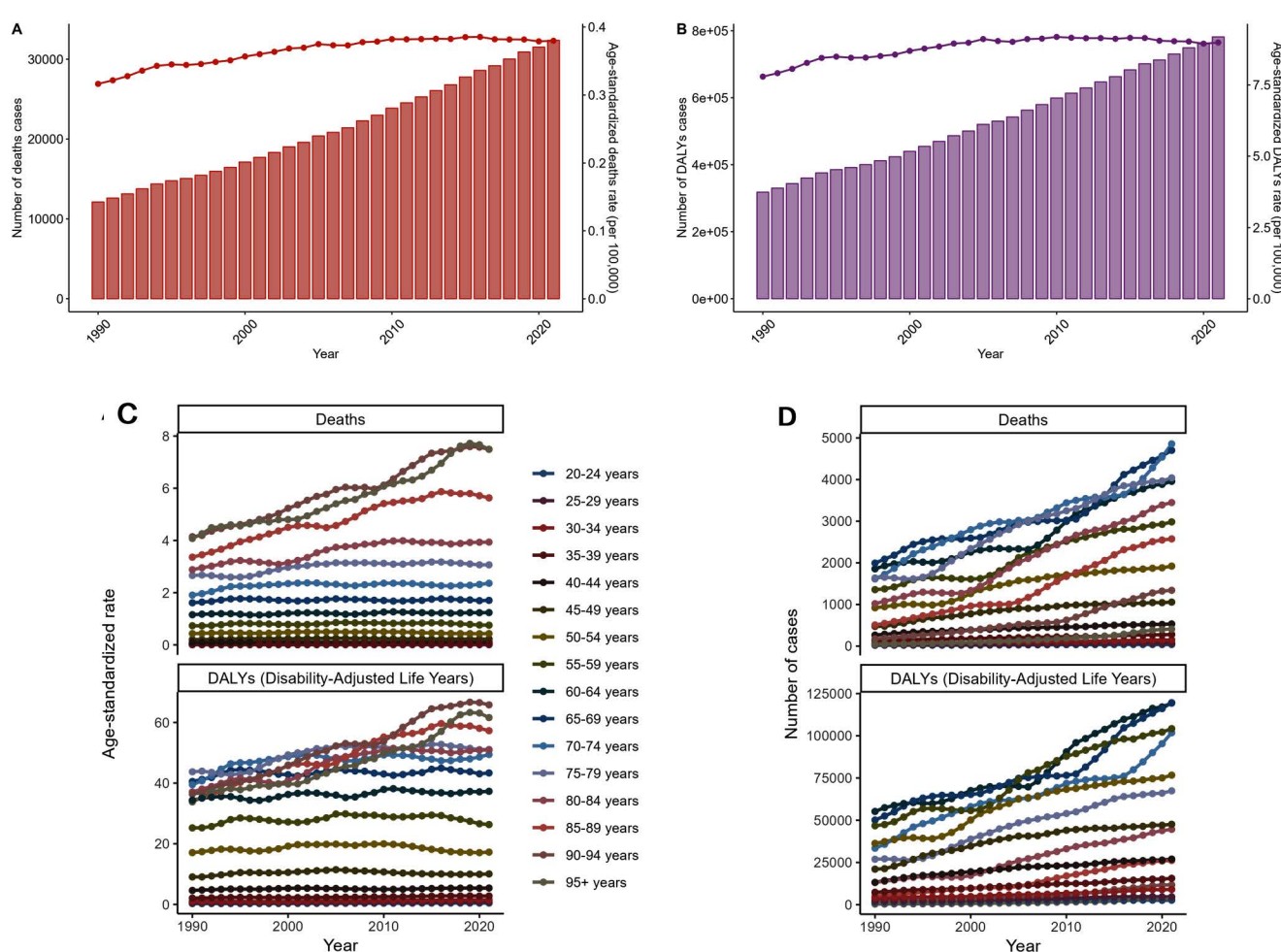

**Fig 4. Trend analysis of health burdens in China and neighboring countries.**

advancements, and shifting geopolitical landscapes, the health metrics of nations are becoming increasingly interdependent, meaning that health issues in one region can quickly become a global concern, necessitating coordinated responses at multiple levels (Fig 4C).

As a key player in the global economy and a populous nation with diverse regional health profiles, China faces unique challenges in maintaining public health [18]. Despite significant advancements in healthcare delivery and technology, certain regions within China are experiencing a disproportionate burden of disease [19]. Factors contributing to this include rapid urbanization, industrialization, lifestyle changes, and environmental influences that exacerbate the prevalence of non-communicable diseases (NCDs) (Fig 4D). To address these trends, policymakers and health authorities must adopt a multifaceted approach aimed at tackling the root causes of rising DALYs and death rates. Public health strategies should encompass preventive measures, such as community-based interventions and public health campaigns designed to reduce risk factors for NCDs. Enhancements in healthcare infrastructure are equally important to ensure timely diagnosis and effective treatment availability. This includes building capacity in primary care, improving access to specialist services, and investing in telemedicine solutions that can reach remote areas (Fig 4D). Furthermore, international collaboration can play a pivotal role in sharing best practices and resources, thereby enabling China to manage these health challenges more effectively [20]. Participation in global health initiatives can facilitate the exchange of knowledge, accelerate research, and secure funding for innovative health programs. Strengthening partnerships with international organizations can also help align domestic policies with global health goals, fostering a cohesive approach to tackling transnational health issues (Fig 4C).

By integrating local, national, and international efforts, it is feasible to mitigate the rising health burdens and contribute to the overall improvement of health outcomes across China [21]. This holistic approach will not only foster a healthier population amidst the dynamic global health landscape but also support sustainable development and economic stability. Through concerted action, we can work towards ensuring that all segments of society benefit from improved health and well-being, regardless of geographical location or socio-economic status (Fig 4D).

## BMI-related health trends

The roles of Body Mass Index (BMI) and the subjective valuation of health have garnered significant attention; however, there is a lack of comprehensive research examining how these factors interact to influence health behaviors [22]. During the course of research on the global burden of high body mass index (BMI) and kidney cancer, a series of graphs labeled as "Fig 5" were encountered, providing valuable insights into the trends of deaths and disability-adjusted life years (DALYs) attributed to high BMI over time. Graph A reveals the increasing number of death cases resulting from high BMI from 1990 to 2020, illuminating the growing challenge faced by various countries worldwide. Graph B complements this observation by presenting the concurrent increase in age-standardized DALYs per 100,000 individuals during the same period. Graph C further explores the issue by comparing deaths and DALYs across different age groups from 1990 to 2020, allowing for analysis of how the burden of high BMI varies across different demographics and informing targeted interventions. Lastly, Graph D investigates the relationship between age-standardized DALYs and the number of deaths, offering a comprehensive view of the impact of high BMI on global health. These graphs serve as vital tools in understanding the complex interaction between high BMI and kidney cancer, enabling identification of patterns, trends, and disparities across countries and age groups. By utilizing this data, evidence-based strategies can be developed to address the mounting burden of kidney cancer associated with high BMI, ultimately aiming to improve global health outcomes.

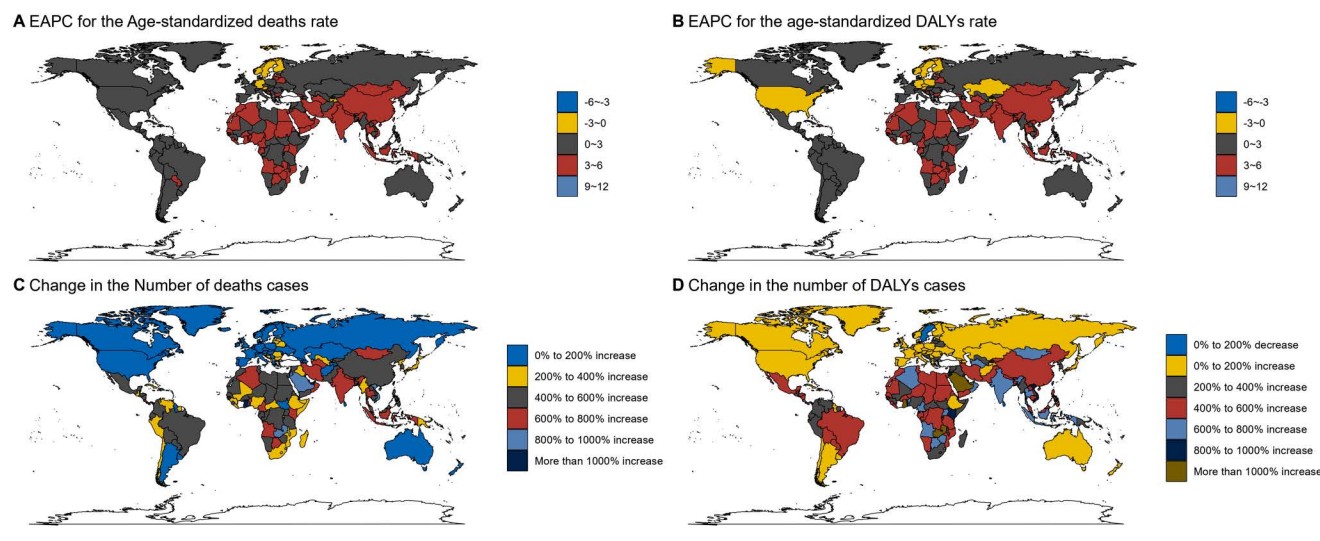

**Fig 5. BMI-Related health trends.**

## Global shifts in BMI-linked renal cancer incidence

Renal cancer incidence is increasing globally [23]. The data presented in Fig 6 unveils critical insights into the temporal trends of a specific disease or health condition, spanning the period from 1990 to 2040. It focuses on deaths, Disability-Adjusted Life Years (DALYs), and age-standardized death rates, providing a comprehensive overview of the disease burden across different demographic and socioeconomic strata. For example, Graph A in Fig 6 illustrates a steady rise in the age-standardized rates of DALYs and deaths, particularly pronounced among lower-income populations. This trend suggests that limited access to healthcare resources and preventive measures might contribute to the disproportionate impact on these groups, underlining the need for equitable healthcare provision. Graph B of Fig 6 indicates a growing number of cases, deaths, and DALYs, signaling an expansion in the disease's prevalence. Future analysis of this data will help us to understand the associations of disease markers with disease relapse and mortality [24]. This trend demands urgent attention from public health stakeholders to implement effective interventions aimed at stemming the tide of this health issue. Moreover, Graph C in Fig 6 reveals a parallel increase in the age-standardized rates of DALYs and deaths for both male and female populations. This highlights the disease's gender-neutral nature and underscores the importance of considering sex-specific risk factors when designing prevention and treatment strategies. Tailored approaches that address the unique needs of each gender could enhance the effectiveness of public health initiatives. Looking ahead, Graph E in Fig 6 projects a substantial rise in fatalities by 2040, emphasizing the urgency of actions aimed at reducing mortality rates and improving survival outcomes. It is imperative that targeted interventions are initiated without delay to mitigate this forecasted increase. Additionally, Graph F of Fig 6 shows an escalating disease burden, as indicated by the rising number of cases and age-standardized DALYs rates. This trend underscores the necessity for proactive interventions to minimize the disease's impact on global health. Researchers, policymakers, and healthcare providers must collaborate in developing evidence-based solutions to curb the spread of the disease and alleviate its consequences. Future studies should delve deeper into potential risk factors, disparities, and effective interventions to combat this health challenge comprehensively [25]. Through concerted efforts and rigorous scientific inquiry, it is feasible to address the multifaceted aspects of this disease and work towards achieving better health outcomes for all populations.

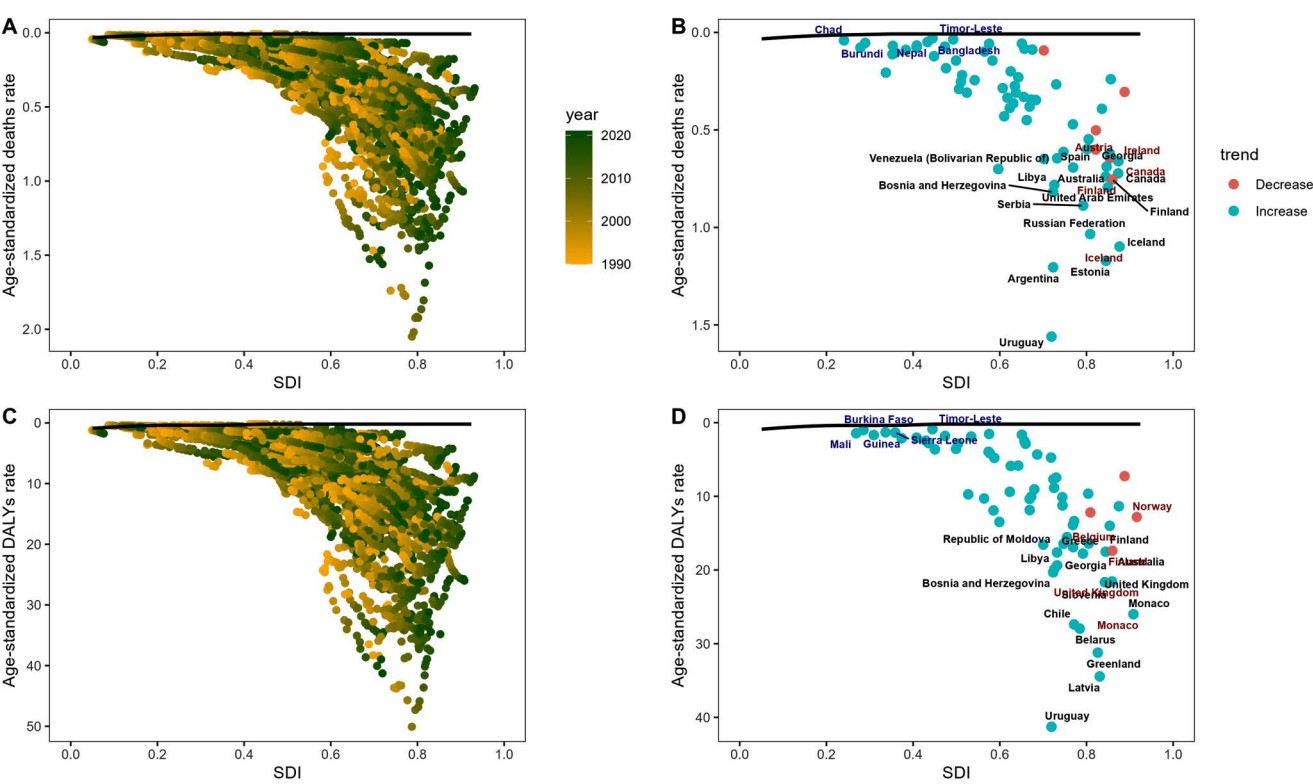

**Fig 6. Global shifts in BMI-linked renal cancer incidence (2025-2040).**

## Global health trends with SDI

The visualizations presented in Fig 7 offer valuable insights into the intricate relationship between age-standardized death rates, age-standardized Disability-Adjusted Life Years (DALYs) rates, and Socio-demographic Index (SDI) values across various countries. By analyzing these relationships, we can gain a deeper understanding of how socio-economic development impacts health outcomes and identify areas for improvement in global health policies. The first and third images in Fig 7 display scatter plots illustrating the correlation between age-standardized death rates and DALYs rates with respect to SDI values. The color gradient in these plots represents the year, allowing us to track the evolution of these rates over time. Typically, countries with higher SDI values—which indicate more developed economies and better social conditions—tend to exhibit lower death rates and DALYs rates. However, there are notable exceptions where countries with moderate to high SDI scores still grapple with reducing their death rates and DALYs rates. This discrepancy could be attributed to a variety of factors, including the quality of healthcare systems, environmental conditions, and unique national circumstances. For instance, countries like Chad, Timor-Leste, and Nepal have experienced an increase in death rates despite having lower SDI values, potentially due to limited access to healthcare resources and inadequate infrastructure. In contrast, countries such as Austria, Ireland, and Finland have seen a decline in death rates, likely attributable to robust healthcare systems and substantial investments in public health initiatives. The second and fourth images in Fig 7 further reinforce these observations by comparing age-standardized death rates and DALYs rates across countries at different SDI levels. The size of the bubbles represents the population, and the trend lines illustrate the overall direction of

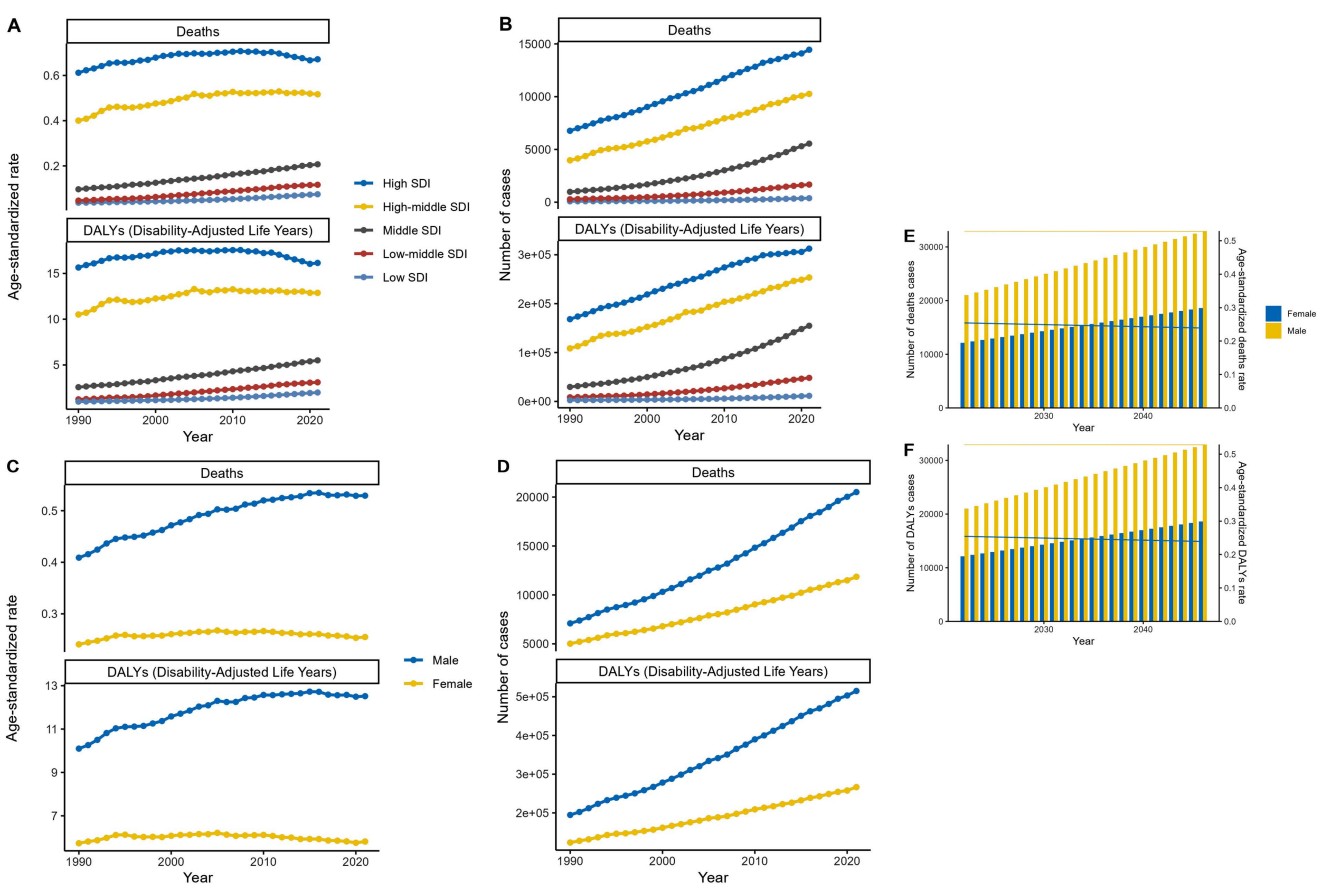

**Fig 7. Global health trends with SDI.**

change in these rates over time. The data suggest that countries with lower SDI values often face greater challenges in reducing their death rates and DALYs rates, whereas those with higher SDI scores tend to enjoy better health outcomes. This emphasizes the importance of investing in healthcare infrastructure and public health programs to improve overall population health. In conclusion, the information presented in Fig 7 reveals the complex interplay between socio-economic development, healthcare infrastructure, and disease burden. While higher SDI countries generally fare better in terms of death rates and DALYs, there are instances where countries with lower SDI values have managed to improve their health outcomes through targeted interventions and investments in healthcare. Conversely, some higher SDI countries face challenges in maintaining or improving their health indicators, possibly due to specific health issues or policy decisions. These findings underscore the importance of continuous monitoring and evaluation of health policies worldwide, ensuring equitable access to healthcare resources and addressing the diverse needs of different populations. Through a concerted effort, we can work towards achieving better health outcomes for all communities, regardless of their socio-economic status.

## Discussion

The results of our study reveal a significant increase in the global incidence of kidney cancer associated with high Body Mass Index (BMI) from 1990 to 2021. By 2021, the incidence

had risen to approximately 720,000 new cases, a marked increase from previous decades. This trend underscores the growing impact of obesity on kidney cancer, a relationship that has been understudied until now [26]. Our analysis indicates that kidney cancer's impact is profound, especially in regions with lower Socio-Demographic Index (SDI) values, such as parts of Africa and South Asia, where despite resource constraints, there have been notable improvements in managing the disease burden. The Estimated Annual Percentage Change (EAPC) method applied to our data set highlights that while there has been modest progress in reducing kidney cancer rates in some regions, other areas continue to struggle. For instance, in low SDI regions, despite an increase in case numbers, age-standardized rates (ASRs) have declined, indicating effective preventive measures and better access to healthcare services. This finding is critical for informing public health policies aimed at reducing the kidney cancer burden in regions that are disproportionately affected.

Our study also addresses the gap in the literature regarding the specific association between kidney cancer and high BMI. By utilizing advanced statistical methods such as Estimated Annual Percentage Change (EAPC) projection, we were able to analyze temporal trends and disentangle the relative contributions of various population factors. These methodologies allowed us to provide a forward-looking perspective on the kidney cancer burden, incorporating contemporary data and insights into the potential impact of global events, such as the COVID-19 pandemic, on kidney cancer prevalence, incidence, and DALYs. However, it is important to recognize that kidney cancer, despite its rising incidence, often receives insufficient attention or treatment, particularly in low- and middle-income countries where research, education, and clinical resources are limited. The findings of this study highlight the need for increased focus on kidney cancer within these contexts, emphasizing the role of lifestyle factors like diet and physical activity in its development.

The data collected from the Global Burden of Disease (GBD) 2021 study offer a nuanced understanding of global kidney cancer trends and underscore the complex interplay between socio-economic conditions and health outcomes. The age and regional distribution of deaths, DALYs, and cases analyzed in this study reveal that while kidney cancer disproportionately affects the elderly, there is also a notable incidence in younger age brackets, albeit with a lower associated DALY count. This suggests that while the prevalence is higher among younger individuals, the severity and overall impact on their health and well-being are less pronounced. Regionally, the disparities in death rates and DALYs rates emphasize the necessity for tailored interventions and strategic resource allocation to regions bearing a heavier kidney cancer burden. The data encapsulated within these analyses equip researchers and policymakers with critical knowledge to inform decisions and develop strategies aimed at addressing the health challenges faced by the most impacted regions and populations. Overall, the study's findings underscore the importance of ongoing investment in public health, regionally adapted interventions, and coordinated efforts among various stakeholders to achieve further reductions in kidney cancer burdens globally. Addressing these trends requires a multi-faceted approach that includes preventive measures, enhancements in healthcare infrastructure, and international collaboration to share best practices and resources. By integrating local, national, and international efforts, it is feasible to mitigate the rising health burdens and contribute to the overall improvement of health outcomes across regions affected by kidney cancer.

## Limitations

Despite the comprehensive nature of our analysis, several limitations should be acknowledged. Firstly, the reliance on GBD 2021 data may introduce biases inherent to the data collection process across different countries. Variations in diagnostic criteria, reporting practices, and healthcare access can affect the accuracy of the kidney cancer incidence reported. Secondly,

the study did not account for all potential confounders that might influence kidney cancer incidence, such as genetic predispositions and environmental exposures. Lastly, while the study provides a global overview, the heterogeneity of kidney cancer trends across regions suggests that localized factors may play a significant role, warranting further investigation in future studies.

## Conclusion

The findings presented herein underscore the importance of addressing the rising incidence of kidney cancer associated with high BMI as a critical global health issue. The study's insights emphasize the need for continued investment in public health infrastructure and the implementation of targeted interventions aimed at reducing obesity rates, thereby potentially mitigating the kidney cancer burden. Furthermore, the analysis highlights the complexity of health disparities across different socio-economic contexts, reinforcing the necessity for equitable healthcare access and tailored strategies to improve health outcomes universally. Through concerted efforts in research, policy-making, and healthcare delivery, it is possible to make strides towards reducing kidney cancer incidence and improving patient outcomes globally.

## Acknowledgments

Thanks to the GBD group for offering us their comprehensive data base.

## Author contributions

**Conceptualization:** Hong Luo.

**Data curation:** Hong Luo.

**Formal analysis:** Hong Luo.

**Funding acquisition:** Hong Luo.

**Investigation:** Hong Luo, Yun Zhao.

**Methodology:** Hong Luo, Yun Zhao.

**Project administration:** Houyu Zhao, Yun Zhao.

**Resources:** Hailiang Jing, Houyu Zhao.

**Software:** Hailiang Jing, Houyu Zhao, Yun Zhao.

**Supervision:** Hailiang Jing, Houyu Zhao, Yun Zhao.

**Validation:** Hailiang Jing, Houyu Zhao, Yun Zhao.

**Visualization:** Hailiang Jing.

**Writing – original draft:** Hailiang Jing.

**Writing – review & editing:** Hailiang Jing.

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
