## [Decision Letter · Decision Letter 0]

6 Jan 2025

PONE-D-24-47755Global, regional, and national burden and trends of kidney cancer associated with high BMI from 1990 to 2021: findings from the Global Burden of Disease Study 2021PLOS ONE

Dear Dr. Zhao,

Thank you for submitting your manuscript to PLOS ONE. After careful consideration, we feel that it has merit but does not fully meet PLOS ONE’s publication criteria as it currently stands. Therefore, we invite you to submit a revised version of the manuscript that addresses the points raised during the review process.

The reviewers commend the manuscript for its comprehensive analysis of kidney cancer epidemiology, leveraging advanced statistical methods and highlighting the growing global burden linked to high BMI. Suggestions include improving table and figure quality and restructuring the discussion into multiple paragraphs for better readability and context.

We look forward to receiving your revised manuscript.

Kind regards,

Wen-Wei Sung, M.D., Ph.D.

Academic Editor

PLOS ONE

2. Please ensure that you have specified a) Did participants provide their written or verbal informed consent to participate in this study?

b) If consent was verbal, please explain i) why written consent was not obtained, ii) how you documented participant consent, and iii) whether the ethics committees/IRB approved this consent procedure."

- In consent please state in Ethics Method section and manuscript if it is written or verbal. If consent was verbal, please explain a) why written consent was not obtained, b) how you documented participant consent, and c) whether the ethics committees/IRB approved this consent procedure.

4. Please ensure that you refer to Figure 2 in your text as, if accepted, production will need this reference to link the reader to the figure.

Additional Editor Comments (if provided):

Reviewers' comments:

Reviewer's Responses to Questions

**Comments to the Author**

1. Is the manuscript technically sound, and do the data support the conclusions?

Reviewer #1: Yes

Reviewer #2: Yes

2. Has the statistical analysis been performed appropriately and rigorously? 

Reviewer #1: Yes

Reviewer #2: Yes

3. Have the authors made all data underlying the findings in their manuscript fully available?

Reviewer #1: Yes

Reviewer #2: No

4. Is the manuscript presented in an intelligible fashion and written in standard English?

Reviewer #1: Yes

Reviewer #2: Yes

5. Review Comments to the Author

Reviewer #1: This paper is well written. The authors aim to provide the most current and comprehensive analysis of kidney cancer epidemiology by utilizing the latest data from the Global Burden of Disease 2021 study and employing advanced statistical methods, such as Estimated Annual Percentage Change projections. The results indicate that the global impact of kidney cancer associated with high BMI has expanded notably from 1990 to 2021, highlighting significant variations across different SDI regions, countries, and sexes. This increasing trend underscores the need for targeted interventions and public health strategies, particularly in regions and populations where kidney cancer prevalence is disproportionately high due to high BMI. These data are helpful in the clinical practice.

Reviewer #2: This manuscript is insightful and important since it is very rich in information and data. However, an improvements in the Table format and the quality of Figures provided in the article might enhance the quality of the manuscript.

Additionally, the discussion section might further be written in paragraphs, instead of using only single paragraph, to enhance the readability and to add more context to every important point stated in the discussion.

6. PLOS authors have the option to publish the peer review history of their article (what does this mean? ). If published, this will include your full peer review and any attached files.

**Do you want your identity to be public for this peer review?** For information about this choice, including consent withdrawal, please see our Privacy Policy .

Reviewer #1: No

Reviewer #2: **Yes: ** Besut Daryanto

---

## [Author Response · Author response to Decision Letter 1]

13 Feb 2025

Dear Editors and Reviewers,

We appreciate the opportunity to revise our manuscript, "Global, regional, and national burden and trends of kidney cancer associated with high BMI from 1990 to 2021: findings from the Global Burden of Disease Study 2021" and thank the reviewers for their insightful comments. These opinions are very valuable for the revision and improvement of our paper, and also have important guiding significance for our research. We provide a detailed response to each reviewer's review. In addition, we have made comprehensive revisions to the entire manuscript and improved the picture quality. We have done our best to write all the revisions clearly and hope that the revised manuscript can meet the requirements for publication.

Sincerely,

Corresponding author.

---

## [Editor Report · Decision Letter 1]

16 Feb 2025

Global, regional, and national burden and trends of kidney cancer associated with high BMI from 1990 to 2021: findings from the Global Burden of Disease Study 2021

PONE-D-24-47755R1

Dear Dr. Zhao,

We’re pleased to inform you that your manuscript has been judged scientifically suitable for publication and will be formally accepted for publication once it meets all outstanding technical requirements.

Kind regards,

Wen-Wei Sung, M.D., Ph.D.

Academic Editor

PLOS ONE
---

## [Editor Report · Acceptance letter]

PONE-D-24-47755R1

PLOS ONE

Dear Dr. Zhao,

I'm pleased to inform you that your manuscript has been deemed suitable for publication in PLOS ONE. Congratulations! Your manuscript is now being handed over to our production team.

Kind regards,

on behalf of

Dr. Wen-Wei Sung

Academic Editor

PLOS ONE